



# Hydrometerological and gravity signals at the Argentine-German Geodetic Observatory (AGGO) in La Plata

Michal Mikolaj[1], Andreas Güntner[1,2], Claudio Brunini[3,4], Hartmut Wziontek[5], Mauricio Gende[4], Stephan Schröder[1], Augusto M. Cassino[3], Alfredo Pasquaré[3], Marvin Reich[1], Anne Hartmann[1], Fernando A. Oreiro[6,7], Jonathan Pendiuk[4], Luis Guarracino[4], and Ezequiel D. Antokoletz[4]

[1]GFZ German Research Centre for Geosciences, Section Hydrology, Potsdam, Germany
[2]University of Potsdam, Institute of Earth and Environmental Science, Potsdam, Germany
[3]Argentinean-German Geodetic Observatory, CONICET, La Plata, Argentina
[4]Facultad de Ciencias Astronómicas y Geofísicas Universidad Nacional de La Plata, La Plata, Argentina
[5]Federal Agency for Cartography and Geodesy (BKG), Leipzig, Germany
[6]Facultad de Ingeniería, Instituto de Geodesia y Geofísica Aplicadas Universidad de Buenos Aires, Buenos Aires, Argentina
[7]Servicio de Hidrografía Naval, Ministerio de Defensa, Argentina

Correspondence: M. Mikolaj (mikolaj@gfz-potsdam.de)

**Abstract.** The Argentine-German Geodetic Observatory (AGGO) is one of the very few sites in the southern hemisphere equipped with a comprehensive cutting-edge geodetic instrumentation. The employed observation techniques are used for a wide range of geophysical applications. The presented multi-compartmental data set provides gravity time series and selected gravity models together with the hydrometeorological monitoring data of the observatory. These parameters are of great inter-
5 est to the scientific community, e.g., for achieving accurate realization of terrestrial and celestial reference frames. Moreover, the availability of the hydrometeorological products is beneficial to inhabitants of the region as they allow for monitoring of environmental changes and natural hazards including extreme events. The hydrological data set is composed of time series of groundwater level, modelled and observed soil moisture content, soil temperature, and physical soil properties and aquifer properties. The meteorological time series include air temperature, humidity, pressure, wind speed, solar radiation, precipita-
10 tion, and derived reference evapotranspiration. These data products are extended by gravity models of hydrological, oceanic, La Plata Estuary, and atmospheric effects. The quality of the provided meteorological time series is tested via comparison to the two closest WMO sites where data is available only in an inferior temporal resolution. The hydrological series are validated by comparing the respective forward-modelled gravity effects to independent gravity observations reduced up to a signal corresponding to local water storage variation. Most of the time series cover the time span between April 2016 and November 2018
with either no, or only few missing data points. The data set is available at https://doi.org/10.5880/GFZ.5.4.2018.001 (Mikolaj et al., 2018).



# 1 Introduction

Existing observation systems at the Argentine-German Geodetic Observatory (AGGO) comprise high-precision geodetic positioning by Global Navigation Satellite Systems (GNSS), Satellite Laser Ranging (SLR), Very Long Baseline Interferometry (VLBI), a high-precision superconducting gravimeter (SG), absolute gravimeters (AG), and seismology. This ranks AGGO among the significant contributors to the global geodetic Earth observation network. Moreover, the authorities committed to a long-term cooperation in providing high-quality data to the international community.

The geodetic observations mentioned above will be or already are distributed via discipline-specific databases such as IGETS (Voigt et al., 2016, igets.u-strasbg.fr, last access 19 November 2018), VLBI IVS/BKG database (www.ccivs.bkg.bund.de, last access 3 December 2018), IGS (www.igs.org, last access 30 November 2018), and SIRGAS (Sánchez et al., 2015, www.sirgas. org, last access 30 November 2018). These databases complement each other, especially owing to the common sensitivity of the observations to Earth's surface displacement. The hydrometeorolgical parameters are essential for large-scale modelling of Earth surface displacement (e.g. Boy and Hinderer, 2006; Dill and Dobslaw, 2013). Local to regional-scale hydrological loadings interfere with variety of geophysical phenomena such as subsidence (e.g. Battaglia et al., 2006; Dixon et al., 2006), preseismic and coseismic changes (e.g. Imanishi et al., 2004; Heki and Matsuo, 2010), or tides (e.g. Braitenberg et al., 2018; Sato et al., 2006). Compared to GNSS, SLR, and VLBI, any gravimeter is additionally sensitive to the direct effect of mass redistribution. Hence, gravity observations can deliver information on surface and sub-surface water storage changes. These include groundwater withdrawals (e.g. Wilson et al., 2011), water recharge (e.g. Kennedy et al., 2016), floods, and storm surges (e.g. Oreiro et al., 2018), all with tangible effect on the inhabitants of the region. These issues gain increasing relevance, given that intense floods causing huge material and partly human losses hit the study region, known as Buenos Aires Pampa, more frequently since 1980. Hence, the availability of comprehensive hydrometeorological and gravity data sets as presented here may contribute to the development of innovative management practices for water resources and natural hazards. In addition, the in-situ hydrological and gravity data are essential for a possible correction of the other geodetic observations on the site and for the evaluation of satellite gravity observations by GRACE and GRACE-Follow On missions using ground-based monitoring (e.g. Crossley et al., 2014; Van Camp et al., 2014).

In this article, we present a data set comprising the majority of the recorded and modelled hydrometeorological and gravity time series at AGGO. The hydrological data set includes soil moisture and groundwater variations. Meteorological time series comprise air temperature, humidity, pressure, wind speed, solar net radiation and precipitation. Additional parameters like soil properties, reference evapotranspiration, local and large-scale gravity models are made available for further use. In this way, the gravity recordings at AGGO can conveniently be reduced for large-scale hydrology, atmosphere and non-tidal ocean loading effects. The data set is divided into three levels comprising observed, processed and modelled time series. Level 1 consists of unmodified recorded data. This type of data is suitable for all users interested in uncorrected observations that are not affected by any processing steps or other data manipulation applied by the provider. Users interested in filtered data corrected for known instrumental issues are advised to use Level 2 products. Level 3 products utilize the Level 2 outputs to model time series such

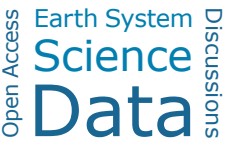

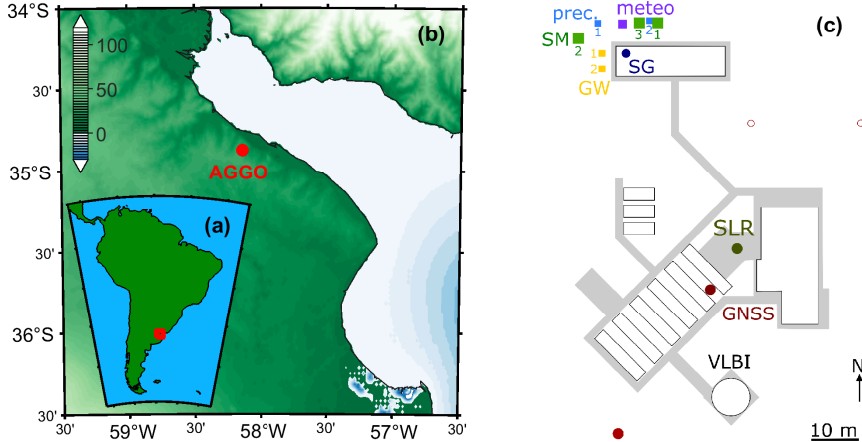

**Figure 1.** Location of the study site (**a** and **b**). The local map (**c** on the right) shows the approximate instrumentation position (in color), buildings as of April 2017 (white), and pavements (gray) at the AGGO site. The map was created using Amante and Eakins (2009); Wessel and Smith (1996) and M_Map toolbox (eoas.ubc.ca/~rich/map.html, access date 2 Noverber 2018)

as evapotranspiration or water storage in the vadose zone. The data set covers approximately two years and a half between April 2016 to November 2018.

## 2 Study site

The Argentine-German Geodetic Observatory was inaugurated in July 2015 as a flagship project of scientific cooperation
between both countries. AGGO is situated north-west of the La Plata city in the Buenos Aires Province (see Figure 1). The topography in the whole area is flat and formed by the sediments of confluencing Parana and Uruguay rivers in the Río de la Plata estuary. The distance of AGGO to the shores of the estuary is approximately 13 km. The estuary width varies significantly and reaches approximately 40 km in the profile crossing the observatory. The proximity to the extremely large estuary plays an important role for observations at AGGO, especially owing to the frequent storm surges. Further details on the charactersitics
of the estuary and its hydrological regimes can be found in Oreiro et al. (2018).

The observatory was constructed on a plane formerly covered by eucalyptus trees. The eucalyptus forest still surrounds the majority of the area of the observatory. There are plans, however, to cut the closest trees which could alter the hydrological regime in the future. The remaining area is covered by grassland, partially used as extensive pasture land. The observatory estate itself is predominately covered by grass with parts filled up with gravel. A geotechnical survey comprising 3 vertical
profiles was carried out prior to the construction of the observatory. All profiles showed clayey soil (soil classification MH) with some calcereous layers up to a spatially varying depth of 3.9 to 6 m. Silty clayey to silty soils (class ML) were found up to the maximum depth of the borehole (10.2 m). The soil samples taken independently of the geotechnical survey for a laboratory analysis are summarized in section 3.1.1 (Table 2).





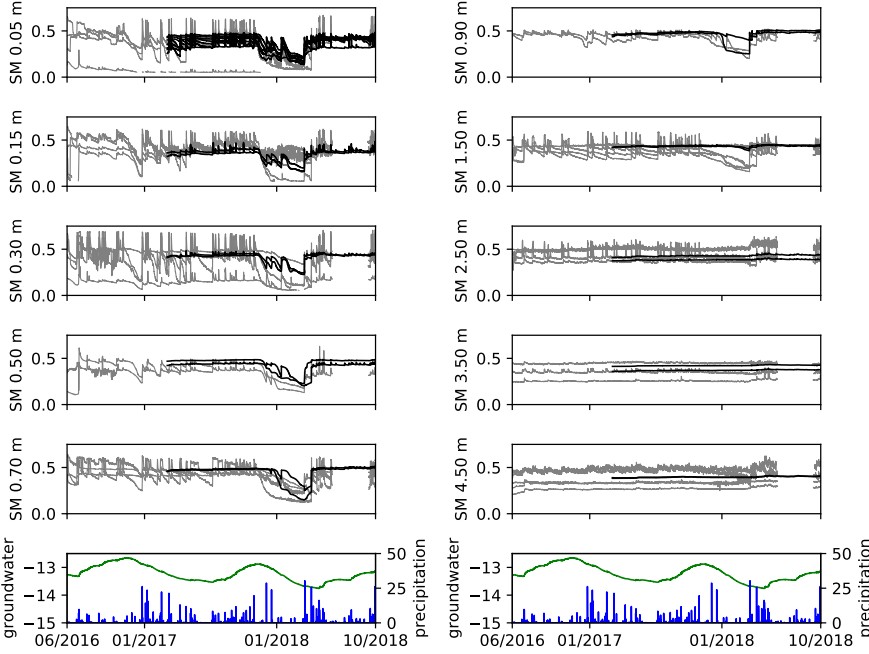

**Figure 2.** All Level 2 soil moisture (SM) series at depths from 0.05 to 4.50 m in $\mathrm{m^3\,m^{-3}}$ units, groundwater (m below surface), and precipitation ($\mathrm{mm\,hour^{-1}}$). Soil moisture recorded with SMT100 sensors are in black, soil moisture recorded with TDR sensors are in gray and additionally filtered using a 13 hour moving window.

The climate at AGGO can be classified according to Kottek et al. (2006) as Cfa (using Koeppen-Geiger climate zone map Rubel et al. (2017)), i.e., humid subtropical climate. The long record from 1961 to 1990 at the meteorological station in La Plata (WMO Station Number 87593) processed by NOAA's National Climatic Data Center (ftp://ftp.atdd.noaa.gov/pub/GCOS/ WMO-Normals, last access 2 November 2018) shows daily mean temperature of 15.8 °C with mean maximum in January

(22.6) and minimum in July (9.2). The mean annual relative humidity equals 77.2% and the mean precipitation reaches 1007 mm. It should be noted that the distance between this meteorological station and AGGO is 24.2 km. Nontheless, similar values (maximal difference of around 4%) are observed at a site north-west of AGGO (36.5 km) in Buenos Aires (WMO Station Number 87576).

From a hydrogeological point of view, AGGO is located over the unconfined Pampeano aquifer (Pleistocene). The Pampeano

formation has a thickness of about 30 m in this area and is composed predominantly of eolian clayey to sandy silt (loess). Underlying the Pampeano is the semiconfined Puelche aquifer (Early Pliocene), which is the main source of groundwater in the region. The Puelche formation is mostly of alluvial origin and it is formed by yellowish quartz sands, with local thin intercalations of gravels and/or clays. The contact between the Pampeano and Puelche formations is often marked by a silty clay layer that confines the Puelche aquifer. The regional groundwater flow of this aquifer system is toward the Río de La Plata

(zone of discharge) with very low hydraulic gradients.





## 3  Data sets

The data set level indicates the degree of data modifications. Level 1 corresponds to the direct observations as collected by the sensors and written by the data logger. Except of the time-domain reflectometry (TDR) measurements, the Level 1 data are aggregations (mean or sum) of 3 previous measurements taken every 5 minutes. Level 2 comprises all Level 1 products after
processing. The first step of the time series processing consists of removing values out of a plausible range. All missing data within a two hour interval were then automatically filled by linear interpolation. Resulting values were used to compute either hourly means (e.g. soil moisture) or hourly sums (precipitation). Known issues or artificial signals were corrected either by interpolation or complete removal, depending on the length of the affected time period. In the last step of Level 2 processing, constant hourly sampling was enforced by flagging missing values. Information about the applied corrections along with system
maintenance records, the local coordinates of the sensors, and installation notes are provided in separate relational tables of the data set.

The modelled data are denoted as Level 3 products. Provided is also the source code and the output of models that were created for this data set. Additional results of other model that were already available for AGGO are included in the data publication as well. Own models include those for evapotranspiration, vadose zone water storage, combined precipitation
series, and gravity effects. Globally available models used for large-scale gravity modelling were also exploited to extract air pressure, temperature, humidity, and water storage variation for the study sites.

### 3.1  Hydrological data

The spatial distribution of the hydrometeorological instrumentation is schematically shown in Figure 1(c). All sensors are located in direct vicinity of the gravimeter building as observations by terrestrial gravimetry are known to be most sensitive to
mass variations in the near-field around the sensors (e.g. Güntner et al., 2017; Reich et al., 2018). Table 1 shows the type and the number of employed hydrological sensors. The accuracy of individual sensors under laboratory conditions can be found for some sensors in manufacturers' specifications (www.campbellsci.com, www.youngusa.com, www.ott.com, www.truebner.de, www.gwrinstruments.com, last access 6 November 2018). Actual accuracy is not provided here as it depends on several varying parameters such as length of sensor cables (e.g., for the TDR system), soil properties, or environmental temperature (e.g., for
the SMT100 sensors). All sensors were deployed utilizing default manufacturer calibration and connected to one of the two data loggers (CR1000 by Campbell Scientific).

### 3.1.1  Soil moisture, temperature, conductivity, and soil properties

A first set of soil moisture and soil electric conductivity sensors was installed at the AGGO site in April 2016. Time-domain reflectometry (TDR) sensors were deployed in 2 soil pits. Each pit was equipped with 2 profiles (SM 1 and 2 in Figure 1(c)
on north and south side of the pit). The manually dug pits allowed for installation of sensors up to a maximum depth of 4.5 meters. 8 (or 10) sensors at 5, 15, 30, (50), 70, (90), 150, 250, 350, 450 cm were distributed in each profile. Photographs of the installation campaign including the pits prior and after installation are part of the data publication. Due to the marked sensitivity



**Table 1.** Hydrological instrumentation at AGGO

| Category | Instrument (manufacturer) | nr. of sensors |
|---|---|---|
| soil moisture | SMT100 (Truebner) | 25 |
| | CS645* 7.5 cm (Campbell Scientific) | 25 |
| | CS635* 15 cut to 5.0 cm probe length (Campbell Scientific) | 15 |
| soil temperature | SMT100 (Truebner) | 25 |
| | CS 107 (Campbell Scientific) | 5 |
| soil electrical conductivity | CS645* 7.5 cm (Campbell Scientific) | 25 |
| | CS635* 15 cut to 5.0 cm probe length (Campbell Scientific) | 15 |
| groundwater level & temperature | OTT PLS (OTT) | 2 |

*used in combination with TDR100 reflectometer and SDM8X50 multiplexer (Campbell Scientific)

of the TDR method to the high electric conductivity of the clayey soil, shortened CS635 sensors had to be used to minimize the travel distance of the electromagentic pulse and to assure sufficient power of the reflected signal. Despite the reduced sensor length, these TDR measurements suffer from high noise, leading to a considerable number of data points out of a physically plausible range. Therefore, a third soil pit with 2 profiles was equipped with SMT100 soil moisture and temperature sensors in

March 2017. These sensors show significantly less noise. Only 0.1% of the SMT100 recordings are missing or are out of range, while almost 14% of the data points recorded by the TDR system had to be discarded. Furthermore, all soil moisture time series should be treated with caution in the first couple of months after installation due to the soil compaction processes going on in the re-filled soil pits in direct vicinity of the sensors. The raw TDR measurements were converted to soil moisture according to Topp et al. (1980). In case of the SMT100 sensors, the provided soil moisture output values relying on the manufacturer's

calibration were directly taken. The soil moisture time series by TDR (gray) and SMT100 (black) sensors are shown in Figure 2.

For characterization of soil-physical parameters, 4 soil samples were taken for laboratory analysis at University of La Plata. All samples were taken from a soil pit that was later used for the gravimeter pillar, i.e., beneath the gravimeter building (location SG in Figure 1(c)). The results of the analysis are shown in Table 2. The lower part of the table shows van Genuchten

parameters estimated with the Rosetta Lite neural network prediction (Schaap et al., 2001) as implemented in the HYDRUS-1D program (pc-progress.com, last access 28 November 2018).

The HYDRUS-1D model (Šimůnek et al., 2016) was set up to quantify water storage variations in the vadose zone between the deepest soil moisture sensor at 4.5 m depth and the groundwater surface. The soil hydraulic properties of the deepest soil sample were used for model parameterization. The upper time-variable boundary condition was set to the pressure head

that corresponded to the mean of all variations observed at 4.5 m depth with the low-noise SMT100 sensors. All missing intervals were linearly interpolated to allow for one continuous model run. The lower boundary pressure head was given by the groundwater level observations described in the proceeding section (3.1.2). The first 3 weeks of the modelled soil moisture were removed to account for the spurious interval related to imperfect initial conditions. The resulting series are denoted as





**Table 2.** Soil physical properties and van Genuchten parameters for four AGGO soil samples at different depths

|  | 30 cm | 100 cm | 200 cm | 380 cm |
|---|---|---|---|---|
| sand (%) | 3.86 | 11.42 | 14.87 | 35.23 |
| silt (%) | 35.22 | 44.29 | 37.38 | 35.33 |
| clay (%) | 60.92 | 44.29 | 47.75 | 29.44 |
| porosity (%) | 42.39 | 49.22 | 50.96 | 42.80 |
| bulk density ($gr\,cm^{-3}$) | 1.25 | 1.30 | 1.28 | 1.43 |
| particle density ($gr\,cm^{-3}$) | 2.17 | 2.56 | 2.61 | 2.50 |
| $Q_r$ | 0.1066 | 0.0981 | 0.0998 | 0.0768 |
| $Q_s$ | 0.5342 | 0.4985 | 0.5055 | 0.4232 |
| $\alpha$ ($m^{-1}$) | 1.86 | 1.30 | 1.51 | 1.13 |
| $n$ | 1.2823 | 1.3871 | 1.3524 | 1.4542 |
| $K_s$ ($cm\,day^{-1}$) | 15.04 | 15.50 | 18.97 | 8.64 |
| $l$ | 0.5 | 0.5 | 0.5 | 0.5 |

Level 3 product sampled every 1.0 m between 5.5 to 11.5, and every 0.2 m between 12.1 and 12.5 m soil depth. Together with the other observation data of soil moisture and groundwater storage, the model output of vadose zone moisture obtained here allows for quantifying total water storage variations at the observatory. This is essential for modelling the gravity signals at the local scale (Section 3.3.2).

### 3.1.2 Groundwater

Two groundwater wells were drilled at the observatory in April 2016 (see GW in Figure 1(c)). The maximum depth of both wells is 33 m with their monitoring filter screen in between 16 and 32 m depth. The groundwater level and temperature observations reflect variations in the uppermost unconfined aquifer at the site, the Pampeano aquifer. The Level 1 groundwater series contain only 0.1% missing values. The Level 2 groundwater level time series were corrected for pump tests and for any missing data points. Linear interpolation could be applied for this purpose due to the minimal noise and the absence of other short-term variations in the Level 1 time series. As shown in Figure 2, a predominantly seasonal signal of groundwater levels can be observed, with an amplitude of about 1 m. The time series of both observation wells are close to identical with correlation $r \approx 1.0$ ($p \approx 0.0$) and a maximum difference of the Level 2 groundwater levels of 1cm. This is related to the small distance of 3 m between both wells, designed for pump test experiments. Groundwater temperature was constant at 17.8°C and no variations that exceeded the precision of the temperature sensor ($\pm 0.5$ °C, www.ott.com, last access 6 November 2018) were observed during the study period.

In order to estimate the specific yield and other hydraulic parameters of Pampeano aquifer a long-term pumping test was performed. The hydraulic test began on 15 May at 1:10 PM and lasted until 17 May 2017 at 20:45 PM. During this period groundwater was pumped at an approximately constant rate of 6.1 m³hour⁻¹ and water levels were measured in the two





monitoring wells. Specific yield values that range from 0.085 to 0.10 were estimated for the Pampeano aquifer using different semi-analytical models implemented in the WTAQ computer program described in Barlow and Moench (1999).

## 3.2 Meteorological data

Table 3 presents an overview of the available meteorological instrumentation. All sensors except for the air pressure sensor are

in operation at AGGO since April 2016. All Level 1 time series show low noise and minimal missing data points equal to 0.2% of the whole provided period (May 2016 to November 2018). The Level 3 products are without any missing data points. The Level 2 meteorological time series discussed in this section are show in Figure 3 (precipitation is shown in Figure 2).

### 3.2.1 Air Temperature, humidity and pressure

The air temperature is recorded by two sensors (see Table 3). Only the CS215 sensor that is also used for relative humidity

measurements is properly shielded against solar radiation. The ambient temperature recorded by the data logger sensor inside an enclosure attached to the pole of the meteorological station should be used only as a proxy in case the CS215 measurements are missing or corrupt. Both measurements are highly correlated ($r = 0.98$, $p \approx 0$). Homogeneity tests carried out using the RHtest software package described in Wang and Feng (2013); Wang (2008a, b) did not disclose any discontinuities (at $\alpha = 0.05$) in either temperature, humidity or pressure.

Unlike other meteorological instrumentation, the atmospheric pressure is recorded by a sensor installed inside the gravimeter building. The instrument was installed at AGGO together with the superconducting gravimeter in December 2015 (Wziontek et al., 2017). Provided here are hourly values starting 1 January 2016 up to November 2018 (1.7% of missing data). The raw source data with one second and one minute resolution can be obtained from the IGETS database hosted at the Information and Data Centre (isdc.gfz-potsdam.de, last access 22 November 2018). The hourly values were linearly interpolated after applying

a low-pass filter with a cutoff frequency of 2.6 hour (at -3 dB) to the one minute data.

Data series aggregated to daily values were compared to those of the two WMO sites that are closet to AGGO (WMO meteorological sites 87576 and 87593, https://www7.ncdc.noaa.gov/CDO/cdo, last access 9 November 2018). Over 99.3% of the variance in temperature series at all three locations can be explained by only one principle component. Similar applies to the air pressure and humidity variation with first component explaining 99.8% and 95.1% respectively. The clear dominance

of large-scale atmospheric processes in the region can be furthermore highlighted by comparison to ERA Interim (Dee et al., 2011) global model. In such comparison, the correlation equals 0.94, 0.86, and 0.99 for temperature, humidity, and pressure, respectively (all $p \approx 0.0$). The acquisition of Level 3 model meteorological series is described in Section 3.3.1.

### 3.2.2 Wind speed, solar radiation, and precipitation

The net radiation at the site can be computed using provided solar shortwave and longwave radiation measured by sensors

facing down- and upward. Radiation data are available in $\mathrm{W\,m^{-2}}$ as 15 minute (Level 1) or 60 minute (Level 2) average. The wind speed is measured at 2 m height but in proximity to a 4 m tall building. Furthermore, the distance to the eucalyptus



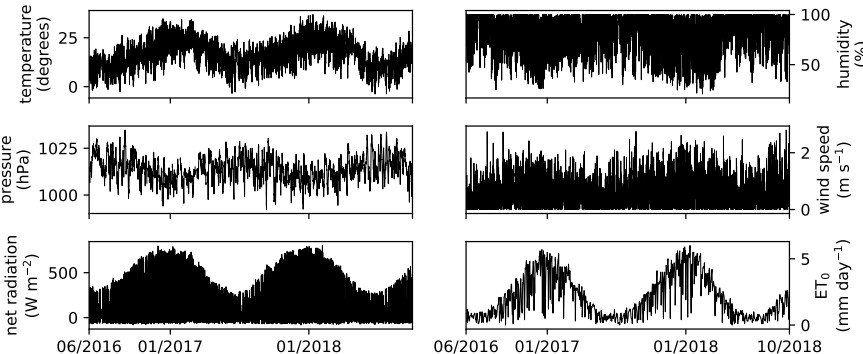

**Figure 3.** Meteorological time series at AGGO

tries is less than 10 m. These obstacles may limit the representativeness of these measurements to a small-scale area only. The correlation computed using daily mean time series of wind speed at AGGO and at the WMO stations 87576 and 87593 equals 0.66 and 0.60, respectively (both $p \approx 0.0$).

The liquid state precipitation at the observatory is recorded by 2 non-heated tipping bucket rain gauges. The distance between
the gauges is 10.9 m, while the shortest distance to building equals 5 m. The distance to the tall eucalyptus tries is around 10 m. Related shielding effects may lead to under-catch of precipitation that is hard to quantify. Moreover, leaves and dirt causes occasional clogging of the instruments. These effects are causing discrepancies between the two time series. A double mass technique disclosed several inhomogeneities. However, the plot of cumulative residuals against time and an associated elipsis at $\alpha = 0.05$ after Allen and Smith (1998) (Annex 4) did not indicate overall inhomogeneity. The regression coefficient equals
0.83 and $r^2 = 0.68$.

A Level 3 continuous precipitation time series was created, addressing the discrepancies between both tipping bucket records. The combination was done manually by revising and replacing values of the first gauge by the second record at time intervals where the discrepancy exceeded 2 mm. In such cases, both WMO sites were used for comparison and for selection of those observations of the two AGGO rain sensors that resulted in closer agreement with the WMO precipitation series. The remaining
missing records were set to zero.

### 3.2.3 Evapotranspiration

The grass reference evapotranspiration ($ET_0$) was computed following the Penman–Monteith FAO-56 standard described in Allen and Smith (1998). Level 2 meteorological data were used as input for the computation. Hourly and daily $ET_0$ estimates were computed separately using constants ($C_d$ and $C_n$) tabulated in Allen et al. (2005). The daily values were checked against
the estimates of the FAO ETo Calculator (www.fao.org/land-water/databases-and-software/eto-calculator/en/, last access 6 November 2018). It should be noted that the aggregated hourly values do not add up exactly to the independently computed daily rates. This is related to the inexact transformation of the equation parameters (e.g., $C_d$) as well as the inherently neglected



**Table 3.** Meteorological instrumentation with approximate height above surface

| Category | Instrument (manufacturer) | nr. of sensors | height (m) |
|---|---|---|---|
| air temperature | CS215 + RAD10 (Campbell Scientific) | 1 | 1.80 |
| | CR1000 (Campbell Scientific) | 1 | 0.80 |
| air humidity | CS215 (Campbell Scientific) | 1 | 1.80 |
| air pressure | Weston 78851C | 1 | 0.80 |
| shor- & long-wave radiation | CNR2 (Campbell Scientific) | 1 | 1.64 |
| wind speed | Wind Monitor Model 05103 (R. M. Young Company) | 1 | 2.00 |
| precipitation | Rain gauge Model 52203 (R. M. Young Company) | 2 | 1.30 |

hourly dynamics when exploiting the daily $ET_0$ equation. For AGGO, the mean difference between aggregated hourly and daily values equals -0.18 $\mathrm{mm\,day^{-1}}$ (95% rounded confidence interval -0.20 to -0.17 $\mathrm{mm}$). Moreover, the null hypothesis of normally distributed differences can be rejected at $\alpha = 0.05$ using Anderson-Darling normality test (Stephens, 1974).

To comply with requirements of most hydrological models for continuous time series, the missing $ET_0$ intervals were filled using $k$-nearest neighbours approach. Minimum and maximum daily temperature, dewpoint and wind speed at WMO La Plata 87593 site were used as proxies. 80% of the computed daily $ET_0$ rates without missing intervals at AGGO were utilized for training. The remaining 20% were used to find $k$ with minimal root-mean-square error of 0.87 $\mathrm{mm\,day^{-1}}$ (not rejecting the null hypothesis of normal distributed errors according to Anderson-Darling test at $\alpha = 0.05$). The predicted daily $ET_0$ rates were equally distributed over missing hourly intervals taking into account computed values if available for part of the affected

day. Prior to the re-distribution, missing intervals over night (9 PM to 6 AM local time) were set to zero automatically.

## 3.3   Gravity

The data set contains gravity residuals as well as modelled gravity variations that aimed at further reduction of the residuals for major environmental effects, such as global atmospheric, oceanic and hydrological mass variations. The modelled series are divided into two main categories, depending on their respective integration radius of mass variations around the site. The local

part refers to gravity effects arising from mass variation within an integration radius of 0.1° spherical distance following the approach of Mikolaj et al. (2015). Nonetheless, the source code provided along with the model outputs allows user to modify the integration radius to any desired value. This applies also for the large-scale effects.

### 3.3.1   Gravity residuals

The observed Level 1 gravity time series of the superconducting gravimeter at AGGO (Wziontek et al., 2017) can be accessed

via IGETS database. The IGETS database also provides Level 2 products processed either by the station operator or at the University of French Polynesia (Voigt et al., 2016). In this study, only Level 3 hourly gravity residuals are provided. The raw gravimeter signal was converted to units of gravity using a calibration factor of -736.5 $\mathrm{nm\,s^{-2}\,V^{-1}}$ and by applying a phase shift of -8.3 seconds. The one second gravity data were subsequently filtered and re-sampled to 1 minute resolution.





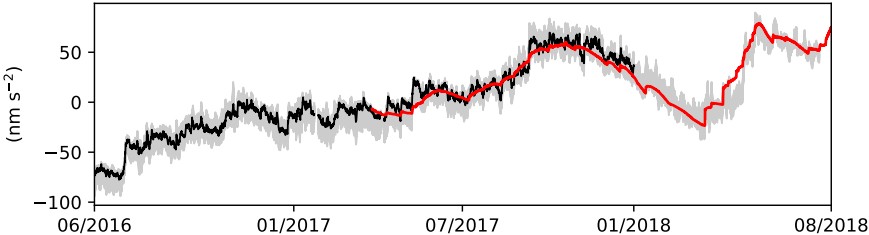

**Figure 4.** Measured gravity residuals reduced for all available large-scale model combinations (in gray). Model combination Atmacs, ODT, and NOAH025 in black. Red line shows the local gravity model.

This gravity time series was then reduced for the effect of Earth and ocean tides applying parameters estimated in a tidal analysis carried out using ETERNA ET34-X-V61 (the updated version V71 is available at ggp.bkg.bund.de/eterna/, last access 26 November 2018). Theoretical tides after Dehant et al. (1999) were used for long-periodic variations (fortnightly and longer). The polar motion and length of day variation was computed using IERS EOP 14 C04 series (datacenter.iers.org, last access: 5

November 2018) after Torge (1989). The instrumental drift equal to $97.72 \pm 3.51 \, \mathrm{nm \, s^{-2} \, year^{-1}}$ was estimated using absolute gravimeter measurements carried out between January and June 2018. Due to the relatively short period between these absolute gravimeter observations, the drift estimate should be used with caution when studying long-term effects. A single admittance approach with $-3 \, \mathrm{nm \, s^{-2} \, hPa^{-1}}$ is used by default to correct the atmospheric effect. However, the residuals can be reduced for the global atmospheric effect discussed in the proceeding section (3.3.2). The gravity time series was furthermore corrected

for steps estimated by visual inspection and corrected for spurious time intervals by means of linear interpolations. Details on these corrections are in metadata tables. Finally, the time series was decimated to hourly temporal resolution by applying the identical low-pass filter as in case of Level 2 atmospheric pressure time series.

### 3.3.2   Large-scale model

The large scale gravity effects are modelled taking into account atmospheric, hydrological and non-tidal ocean mass transport.

All hydrological effects are computed using mGlobe toolbox described in Mikolaj et al. (2016). The input model data are listed in Table 4. The gravity effects were computed for integration radius greater than 0.1°. The enforcement of mass conservation was implemented by applying a uniform layer over the ocean. The gravity response to such variation was computed assuming equal redistribution of model mass deficit or surplus compared to long-term mean. This approach did not take the the mostly unreliable storage estimations over Antarctica and Greenland (set to zero). The global hydrological models were also exploited

to obtain the Level 3 total water storage variations. It should be noted that non of these input models covers the whole saturated and unsaturated zone and should therefore be used accordingly.

The atmospheric effect was computed using three different input models. ERA Interim was used in combination with mGlobe toolbox (Mikolaj et al., 2016). The gravity effect corresponding to mass transport as modelled by ECMWF Operational were directly obtained from EOST Loading Service (loading.u-strasbg.fr, last access 8 October 2018). Similar applies to the ICON





384 global atmospheric model that is utilized in the Atmacs service (atmacs.bkg.bund.de, last access 8 October 2018). In addition to the atmospheric gravity effect, the model surface air pressure, humidity and temperature were extracted to be used in the database (Level 3 products). For ERA Interim, the time series were obtained using simple spatial linear interpolation. Atmacs provides only the model pressure at AGGO without need for spatial interpolation. In case of the EOST products, the

pressure time series were obtained after dividing the local contribution by a given conversion factor. The model pressure should be used in combination with in-situ observations to refine the total atmospheric gravity effect as described in Mikolaj et al. (2016).

To effect of non-tidal ocean loading to gravity variations at AGGO was computed using four models with global coverage. The ECCO1 (ECCO-JPL), ECCO2 and TUGOm gravity effects were downloaded from the EOST Loading Service (loading.

u-strasbg.fr, last access 8 October 2018). Additionally, the effect was computed by utilizing OMCT RL06 model in combination with mGlobe toolbox. The non-tidal loading effect of the Río de La Plata Estuary was modelled after Oreiro et al. (2018). Like in the case of hydrological and atmospheric effects, the full spatial resolution of all input models was used for the computation.

The gravity effect was computed for all components using the highest available temporal resolution. The only exception was the hydrological effect where daily data were used. This simplification has a minimal effect on gravity as shown in Mikolaj

et al. (2016). Hourly Level 3 time series provided in the database were obtained after linear interpolation. The comprehensive set of large-scale gravity effects allows for computation of gravity residuals reduced for global hydrological, atmospheric and oceanic signals including minimum-maximum bounds. These bounds can be estimated by reducing the gravity residual for all possible combinations of available model. This approach presumes that the true large-scale gravity effect is not known and each model is treated as equally accurate. The result is shown in Figure 4. In black are the residuals reduced for one particular

model combination of large-scale effects, namely NOAH025, Atmacs, and ODT model. The latter model was chosen because of the efficient reduction of gravity effects of storm surges in the La Plata estuary. The residuals reduced for the large-scale gravity effects using all other model combinations are shown in gray (105 combinations in total).

### 3.3.3 Local model

The local model of the water storage variations in the subsurface of the observatory extends the large-scale hydrological gravity

models described in the previous section. Therefore, the local effect is computed for the whole area up to the integration radius of 0.1°. However, in view of the minimal altitude variations in the study region and, thus, an assumption of a flat topography, only mass variations within approximately 100 m around the site efficiently contribute to the gravity effect (e.g. Güntner et al., 2017). In addition, soil moisture variations directly bellow the footprint of gravimeter building were set to zero in accordance with Reich et al. (2018). Vertical discretization was set to fit the depth of actual soil moisture measurements, i.e., with first

layer between 0.0 to 0.1 up to the last layer between 4.0 to 5.0 m (see Section 3.1.1). A prism approximation was used for this purpose (Banerjee and Gupta, 1977). The low-noise Level 2 soil moisture time series collected by SMT100 sensors were used to compute the time-variable local gravity effect. The effect of groundwater storage variations was estimated by converting the groundwater level time series with a specific yield equal to 0.1 (10 %) as estimated in the pump test. The gravity effect of the vadose zone between the lowest soil moisture sensor and the groundwater level was quantified using the



**Table 4.** Large-scale gravity models for atmospheric (atmo), hydrological (hydro), non-tidal ocean (ntol) and estuary loading effects

| Model | | Reference | | Data |
|---|---|---|---|---|
| Name | Type | Input data | Processing | Access[*] |
| GLDAS/CLM | hydro | Rodell et al. (2004) | Mikolaj et al. (2016) | disc.gsfc.nasa.gov |
| GLDAS/MOS | hydro | Rodell et al. (2004) | Mikolaj et al. (2016) | disc.gsfc.nasa.gov |
| GLDAS/NOAH025 (v21) | hydro | Rodell et al. (2004) | Mikolaj et al. (2016) | disc.gsfc.nasa.gov |
| GLDAS/VIC | hydro | Rodell et al. (2004) | Mikolaj et al. (2016) | disc.gsfc.nasa.gov |
| ERA Interim | hydro, atmo | Dee et al. (2011) | Mikolaj et al. (2016) | apps.ecmwf.int |
| MERRA Reanalysis 2 | hydro | Gelaro et al. (2017) | Mikolaj et al. (2016) | disc.gsfc.nasa.gov |
| NCEP Reanlysis 2 | hydro | Kanamitsu et al. (2002) | Mikolaj et al. (2016) | esrl.noaa.gov |
| ICON 384 | atmo | Zängl et al. (2014) | Klügel and Wziontek (2009) | atmacs.bkg.bund.de |
| ECMWF operational | atmo | | Boy et al. (2009) | loading.u-strasbg.fr |
| ECCO1 | ntol | Fukumori (2002) | Boy et al. (2009) | loading.u-strasbg.fr |
| ECCO2 | ntol | Menemenlis et al. (2008) | Boy et al. (2009) | loading.u-strasbg.fr |
| TUGOm | ntol | Loren and Florent (2003) | Boy et al. (2009) | loading.u-strasbg.fr |
| OMCT RL06 | ntol | Dobslaw et al. (2017) | Mikolaj et al. (2016) | ftp://isdcftp.gfz-potsdam.de |
| ODT | estuary | Oreiro et al. (2018) | Oreiro et al. (2018) | |

[*]*Last access: 8 October 2018*

local hydrological model (HYDRUS-1D) described in section 3.1.1. Resulting Level 3 local gravity effect time series were computed as the sum of all storage compartments, i.e., observation-based soil moisture, observation-constrained simulated vadose zone water storage, and observation-based groundwater storage variations. This composition allows for an independent validation of the hydrological products by comparing the result of the local gravity model to gravity residuals. As mentioned

in the previous section, the gravity residuals need to be further reduced to signal corresponding to local hydrology by applying the aforementioned large-scale effects. The resulting gravity variations for one particular combination (NOAH025, Atmacs ODT) of large-scale effects is shown in Figure 4 in black, while all other possible combinaitons are shown in gray. The red thick line corresponds to the local hydrological effect discussed in this section. The close correspondence of the resulting gravity residuals with the local hydrological gravity effect proves, on the one hand, the quality of the multi-compartmental

data sets for gravity reductions based on local and global observations and models, and, on the other hand, the quality of the hydrometeorological monitoring system and its data set provided here for assessing the hydrological dynamics at AGGO.

## 4 Data and Code availability

The data set (Mikolaj et al., 2018, https://doi.org/10.5880/GFZ.5.4.2018.001) and code associated to the processing and modelling of the data (Mikolaj, 2018, https://doi.org/10.5880/GFZ.5.4.2018.002) are published via GFZ Data Services. The data

set is organized in a database structure and prepared for implemented in a relational database. Nevertheless, all definitions and





data tables are provided in separate text files allowing access without need for installation of a management system. However, the use of the relational database is advisable as it allows for easy access to all metadata information such as installation notes, sensor types, or applied reductions.

## 5 Conclusions

This study presents hydrological, meteorological and gravity time series observed and modelled at the Argentine-German Geodetic Observatory (AGGO) between April 2016 and November 2018. Thanks to the existing and maintained infrastructure, the data set can be extended in the future to allow for studies of long-term variability and trends. Raw uncorrected, processed, and modelled series denoted as Level 1, 2, 3 products, respectively, are provided. The directly observed series are suitable for users interested in observations that are not affected by any filtering and subjective data manipulation. Level 2 comprises

time series corrected for instrumental and other issues while applying unified processing standards. The modelled series are tailored for studies where continuous homogenized inputs are needed. These may include hydrological modelling for water management or research purposes, verification of meterological models, or use of gravity observations for interpretation of local geophysical phenomena. The gravity models are also of interest for studies aiming at evaluation of Gravity Recovery and Climate Experiment-Follow On satellite mission via inter-comparisons to terrestrial observations. Furthermore, the presented

data set directly feeds into the contributions of the AGGO observatory to realization and maintenance of regional to global-scale terrestrial reference frames. The adequate consideration of local hydrological effects and loading-induced variations as provided in study is required for this purpose.

*Author contributions.* MM drafted and coordinated the work on the manuscript, processed and modelled the time series, and compiled the database. AG, MR and MM designed the hydrometeorological monitoring network at AGGO. AH, EA, HW, FO contributed to data

modelling. LG, JP conducted and evaluated the pump test. AP, AC, SS, AG, MR, CB participated in instrument installation and contributed to the station maintenance. AG, CB, MG and HW acquired project funds, supervised the project and revised the manuscript.

*Competing interests.* The authors declare that they have no conflict of interest.

*Acknowledgements.* The compilation, processing and publishing of the data set was carried out within the Hydrological and oceanic signals in geodetic observations at the Argentine-German Geodetic Observatory (HOSGO) project (number 01DN16019) funded jointly by Bun-

desministerium für Bildung und Forschung (BMBF) and Consejo Nacional de Investigaciones Científicas y Técnicas (CONICET). Authors thank the whole AGGO staff for great support while setting up and maintaining the instrumentation. We also thank Mehedi Hasan (GFZ Potsdam) for helping with the SQL databases, Yang Feng for provision of the RHtest, and Klaus Schueller for further developement of the ETERNA package. We are deeply indebted to the Julia, Octave/Matlab and R developer community. Observed gravity time series and OMCT

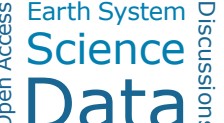

data used in this study was obtained from Information System and Data Center for geoscientific data servers at GFZ Potsdam. Selected gravity models were obtained from EOST Loading, and Atmacs services. The GLDAS and MERRA data used in this was were acquired as part of the mission of NASA's Earth Science Division and archived and distributed by the Goddard Earth Sciences (GES) Data and Information Services Center (DISC). The ERA-Interim data used in this study was obtained from the ECMWF data server. The NCEP Reanalysis data
5   was provided by the Physical Sciences Division of NOAA/ESRL.





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
