# Peer review of "Hydrometeorological and gravity signals at the Argentine-German Geodetic Observatory (AGGO) in La Plata"

_Earth System Science Data, 2018_

## Referee Comment (RC1) · Anonymous Referee #1 · 25 Feb 2019

This paper provides an overview of Level 1, 2, and 3 data products from AGGO. The paper is well written and the analysis thorough. The data are likely to be useful to researchers in several fields. The publication, including code and documentation, provides a very good template/example for similar datasets. The authors have collected an impressive dataset that appears to capture essentially all of the contributions to the gravity signal. Although I am interested in further analysis of the hydrological signal, that's probably beyond the scope of this paper. I don't have any major comments about this relatively straightforward publication.

The authors have done a very nice job providing code used in data preparation and

documentation of the code and data. I have downloaded and briefly explored the accompanying data sets and code but have not thoroughly reviewed their accuracy or the quality of the processing. It would be useful for casual inspection if the authors could include a sub-sampled (daily?) dataset in Tsoft format.

Minor suggestions and corrections:

Title: "Hydrometerological" should be "Hydrometeorological"

P1 L2: should be "equipped with comprehensive . . ."

P1 L3: "multi-compartmental" is an odd descriptor for a data set

P2 L7+: Suggest stating which data is stored in which database.

P2, L11: Change "parameters" to "observations", for consistency with line 7? I found this paragraph to be somewhat disjointed, i.e., it bounces around between a few different ideas.

P2, L27: I tend to think of model parameters when I hear parameters, but I think you are referring to observations and modeled gravity time-series. Are "local and large-scale gravity models" a parameter?

P2, L32: I would add a sentence that explicitly states what level 2 data re, e.g., "Level 2 data consist of level 1 data corrected for artefacts and gaps in the data. . ."

P3: Suggest adding the specific coordinates of the site. I was interested in seeing a satellite image but was unable to locate it using the information in Figure 1 and the text.

P3 L11: plain not plane

P4 L15: It seems you are indicating groundwater flow is to the NW, parallel to the coast and opposite the direction of flow in the Rio de La Plata? Unusual.

P5: Suggest including the time interval at which data sets are reported.

P5, L7: How were data gaps longer than 2 hours handled?

P5, L14: "Own models" is awkward phrasing; suggest "Models developed for this study were those for. . ." or similar.

P5: I realize reporting uncertainty for each measurement is a large undertaking, but it would be helpful to have some idea of the relative uncertainties of each component. Its not necessarily within the scope of the paper.

P5 L29: SM1 and 2 refer to the soil pits, not the profiles, correct? Deep pits! "Manually dug" would imply shovels, not heavy machinery.

P6 L10: Is the mfg.'s calibration specific to the soil type? It looks like the SMT100 probes output permittivity – is it useful to compare the mfg. calibration to the Topp equation?

P6 L18: suggest replacing groundwater surface with water table, and including the depth to water.

P7: I would mention that groundwater levels were recorded with submersible pressure transducers.

P7 L7: a screen interval to 32 m depth would place it below the 30-m thick Pampeano formation (P4 L10). Can you state that the wells didn't penetrate the Puelche formation, or that the groundwater levels are a composite of the two formations? If the intent is to measure gw levels in the Pampeano, its surprising they would be screened with such a long interval, and so close to the bottom of the formation.

P7 L13: Its unclear what p is here and elsewhere. The p-value from a statistical test?

P8 L8: These SY values appear to agree very well with the gravity data, based on figure 4. At some point it would be interesting to compare those estimates, not necessarily in this paper. But you could mention the good agreement (some readers may not realize gravity data are useful for estimating SY).

P8 L17: delete "of" (1.7% missing data).

P8, L18: I only found data through March 18 in IGETS. I assume it will be updated at some point? Assuming this is a long-term site, can these data sets (from this paper and IGETS) be maintained/updated "automatically"?

P8 L21: Suggest defining "WMO" abbreviation at first use

P8 L27: Section 3.3.1 describes gravity residuals, do you mean 3.3.2 and/or 3.3.3?

P9 L1: trees not tries

P10 L12: I would state explicitly what corrections were applied, e.g. "The data set contains gravity residuals corrected for. . .as well as. . ."

P10 L20: Are Level 1, 2, 3 in this paper used the same as at IGETS? That would be worth mentioning in the introduction.

P10 L19: Here you discuss gravity time series under the heading "Gravity residuals". Maybe move the mention of Level 1 and level 2 to the general "Gravity" heading?

P10 L 21: "In this study, only Level 3 hourly gravity residuals are provided": unclear. Do you mean, gravity residuals are only provided as a level 3 product? (Do IGETS Level 2 products include residuals?)

P10 L 22: I would be interested to learn how the calibration factor was determined.

P12 L8: You could mention storm surges here as a major contributor to the non-tidal ocean loading – it took me a while (and the Oreiro paper) before I figure out what this was.

P12 L10: what exactly is the hydrological effect? (soil moisture + groundwater + precip + ET?) Its surprising daily rainfall would suffice for hourly residuals.

P12 L 30: Add "m" after 0.1

P13: I would mention specifically that code is provided as Matlab, Julia, and shell scripts (+ others?), and that the relational database is SQL. For what it's worth, I was

unable to follow the instructions for using MySQL (I don't have any experience using it). I was able to create a database and run the commands in create_hosgo_db.sql and fill_hosgo_db_metadata.sql from the SQL command prompt, but I got several errors trying to run the commands in fill_hosgo_db_data.sql, all of the form:

ERROR 1452 (23000): Cannot add or update a child row: a foreign key constraint fails ('hosgo'.'timeseries', CONSTRAINT 'timeseries_ibfk_1' FOREIGN KEY ('ts_id') REFERENCES 'timeseries_info' ('ts_id'))

Figure 1: Label elevation scale bar in meters. A satellite image in part (c) would be useful.

Figure 2: Be more specific about groundwater units, both in the y-axis label and the caption. If you are reporting negative values, it is probably groundwater elevation relative to land surface. More typical would be "Depth below land surface", with positive values and a reverse y-axis, or elevation relative to mean sea level, also in positive values.

Figure 4: Perhaps outside the scope of the paper, but I would be interested to see additional time series: the gravity effect of soil moisture, goundwater, air pressure, local loading, global loading, etc., plus the gravity residuals before applying air pressure and hydromet corrections. It appears you've simulated the residuals nearly exactly from the hydromet data and models; what does the residual look like after that correction – it must be nearly flat? What signal(s) might you see in such a time series?

References: There appears to be a formatting error in which the URL is duplicated (with slight changes) for many of the references.

---

## Referee Comment (RC2) · Anonymous Referee #2 · 13 May 2019

In 2015, after moving to La Plata in Argentina, TIGO became AGGO - the Argentine-German Geodetic Observatory. The present manuscript shows that the observatory has started its work in full and with great success. The manuscript is written fluently and provides a good overview of the data sets provided, their quality and possible interpretations - although the latter is not the subject of the manuscript. Depending on the degree of processing, the data from pedology/geology, meteorology and geophysics/geodesy are clearly presented in three categories: 1 - raw data, 2 - processed and 3 - user-friendly.

The manuscript describes the instrumentation of the observatory, the processing of the

raw data and the results. These make you eager for results when the time series has become longer. I didn't see any big deficits in the presentation and organization of the manuscript and had a lot of pleasure working through it.

In the following I would like to note a few minor details. (1) First there are slight redundancies in the representation of the 3 data levels (page 2, line 30 and following) and p. 5, L 2 and following. (2) Please do not use cgs units but SI units (p. 7 table 2) (3) The accuracy of the percentages (in the first column in Table 2) allows a number representation up to the second decimal? (4) In general, I find the spatial relationship between illustrations and description in the text to be too large. Both should be presented more in relation to each other. The same applies to the tables (Table 4 and Section 3.3.2).

Figures Please, show in fig. 1a the position of the cities of La Plata and Bs. Aires. Enlarge fig. 1b and replace the yellow colour with a different one – it is hard to read. Explain "prec." "meteo", SM (??), SLR, GNSS etc. I suggest to include a photo showing some parts of the interior – if possible.

All other pictures are too small for my opinion - enlarge, if possible.

---

## Referee Comment (RC3) · Jeff Freymueller (Referee) · 14 May 2019

My comments are limited to minor corrections, as shown in the annotated manuscript. The data set looks to be complete and useful, and the descriptions are comprehensive.

Please ignore Fig. 1, a track changes version has been uploaded as supplement.

Please also note the supplement to this comment:
https://www.earth-syst-sci-data-discuss.net/essd-2018-156/essd-2018-156-RC3-supplement.pdf

[revised manuscript text omitted]

---

## Author Comment (AC1) · 5 Jul 2019

**Reply to referees' comments of the manuscript Hydrometeorological and gravity signals at the Argentine-German Geodetic Observatory (AGGO) in La Plata**

We are grateful to all referees for their careful review of our manscript and for their positive and constructive comments and suggestions. In the following, we reply to all of them in a point-by-point response. The referees' comments are given in italic, the authors' responses are in regular font.

Reply to "**RC1 by Anonymous Referee 1**"

We thank the referee for his very positive overall evaluation of the manuscript. Here are our answers to his/her specific comments:

*It would be useful for casual inspection if the authors could include a sub-sampled (daily?) dataset in Tsoft format*

– For the purposes of a casual inspection, the data repository contains plots of each time series. Although these plots are not interactive, users can quickly gain the basic information on all parameters and products (levels) without the need to install or load any additional files. The corresponding PNG files are located in the docu/plot/all_series folder.

*Title: "Hydrometerological" should be "Hydrometeorological"*

– Spelling corrected in the revised manuscript.

*P1 L2: should be "equipped with comprehensive..."*

– Corrected.

*P1 L3: "multi-compartmental" is an odd descriptor for a data set*

– Modified: "The presented data set provides gravity time series and selected gravity models together with the hydrometeorological monitoring data of the observatory."

*P2 L7+: Suggest stating which data is stored in which database*

– Modified: "The geodetic observations mentioned above will be or already are distributed via discipline-specific databases such as IGETS for SG (igets.u-strasbg.fr, last access 19 November 2018), VLBI IVS/BKG database (www.ccivs.bkg. bund.de, last access 3 December 2018), IGS (www.igs.org, last access 30 November 2018), and SIRGAS (www.sirgas. org, last access 30 November 2018), both storing GNSS observations."

*P2, L11: Change "parameters" to "observations", for consistency with line 7? I found this paragraph to be somewhat disjointed, i.e., it bounces around between a few different ideas*

– Yes, we agree, "parameters" replaced by "observations".

– The paragraph is meant to outline the different but related aspects that motivated this study and the collection of the data sets presented here. Given the reviewer's comment, we re-arrange the paragraph to make the line of thoughts more fluent.
In the revised manuscript, the paragraph reads (references omitted here): "The geodetic observations mentioned above will be or already are distributed via discipline-specific databases such as IGETS for SG, VLBI IVS/BKG database, IGS, and SIRGAS, both storing GNSS observations. These databases complement each other, especially owing to the common sensitivity of the observations to Earth's surface displacement. Surface displacements are caused by a variety of geophysical phenomena such as subsidence, pre-seismic and co-seismic changes, tides, or local to regional-scale hydrological loadings due to water storage changes. Hydrometeorolgical observations such as those presented in this study are essential for modelling of these Earth surface displacements. Compared to GNSS, SLR, and VLBI, gravimeters

are additionally sensitive to the direct effect of mass redistribution. Hence, gravity observations can deliver information on surface and sub-surface water storage changes. These include groundwater withdrawals, water recharge, floods, and storm surges. Such processes and events may all have tangible effects and increasing relevance for the inhabitants of the study region, known as Buenos Aires Pampa, given that intense floods causing huge material and partly human losses hit the area more frequently since 1980. Hence, the availability of comprehensive hydrometeorological and gravity data sets as presented here may contribute to the development of innovative management practices for water resources and natural hazards. In addition, the in-situ hydrological and gravity data are essential for correcting the other geodetic observations of the observatory for hydrological effects so that they may be more suitable for studying other geophysical processes such as those mentioned above, and for the evaluation of satellite gravity observations by GRACE and GRACE-Follow On missions using ground-based monitoring."

*P2, L27: I tend to think of model parameters when I hear parameters, but I think you are referring to observations and modeled gravity time-series. Are "local and large-scale gravity models" a parameter?*

– We agree with the reviewer in using the term "parameters" only in the context of model parameters. In this section, the parameters refer to soil properties only. We thus modified the sentence to: "Additional modelled variables and parameters like soil properties, reference evapotranspiration, local and large-scale gravity time series are made available for further use."

*P2, L32: I would add a sentence that explicitly states what level 2 data are, e.g., "Level 2 data consist of level 1 data corrected for artefacts and gaps in the data. . .*

– A more explicit description has been added: "Level 2 data consist of Level 1 data corrected for artefacts and gaps. "

*P3: Suggest adding the specific coordinates of the site. I was interested in seeing a satellite image but was unable to locate it using the information in Figure 1 and the text*

– The coordinates ($\phi = 34^o52'24''$S, $\lambda = 58^o8'24''$W) have been added to the figure description.

*P3 L11: plain not plane*

– Corrected.

*P4 L15: It seems you are indicating groundwater flow is to the NW, parallel to the coast and opposite the direction of flow in the Rio de La Plata? Unusual.*

– The groundwater flow direction is towards the Rio de La Plata estuary. Thus, it is to the North East and about perpendicular to the coast. We add the term 'estuary' in the revised version to make this clear. Thanks for the hint.

*P5: Suggest including the time interval at which data sets are reported.*

– We added to the Data sets section: "The maximal temporal coverage of the data set ranges from May 2016 up to November 2018 with some exceptions for sensors and models set up in May 2017". More details, including the time resolution, are given in the specific descriptions of each variable / data set.

*P5, L7: How were data gaps longer than 2 hours handled?*

– Added: "If not stated otherwise (e.g., Groundwater section), longer gaps were not filled."

*P5, L14: "Own models" is awkward phrasing; suggest "Models developed for this study were those for. . . " or similar.*

– Modified accordingly.

*P5: I realize reporting uncertainty for each measurement is a large undertaking, but it would be helpful to have some idea of the relative uncertainties of each component. Its not necessarily within the scope of the paper.*

– As noted by the referee, a comprehensive uncertainty requires a significant additional effort which is beyond the scope of this paper. Therefore, in the individual data sets descritpions, we refer to the uncertainties as provided by the manufacturers of individual sensors. In addition, the following reference on uncertainty analysis of gravity corrections (models) is added in section Large-scale model in the revised mansucript:

5   – Mikolaj et al., (2019) "Resolving Geophysical Signals by Terrestrial Gravimetry: A Time Domain Assessment of the Correction-Induced Uncertainty", JGR-Solid Earth, https://doi.org/10.1029/2018JB016682

*P5 L29: SM1 and 2 refer to the soil pits, not the profiles, correct? Deep pits! "Manually dug" would imply shovels, not heavy machinery.*

– Correct, SM1 and 2 in Figure 1 are soil pits with profiles on 2 opposite sides of the pit. It is amazing, but these deep pits
10    were in fact dug manually, with shovels only, by local workers who deserve a lot of respect.

*P6 L10: Is the mfg.'s calibration specific to the soil type? It looks like the SMT100 probes output permittivity – is it useful to compare the mfg. calibration to the Topp equation?*

– As mentioned in the Hydrological data section, "all sensors were deployed utilizing default manufacturer calibration and connected to one of the two data loggers". This also applies to the SMT100 sensors where the sensor output in volumetric
15    water content is directly taken. A soil-specific calibration of the sensors has not been performed.

*P6 L18: suggest replacing groundwater surface with water table, and including the depth to water.*

– Replaced: "groundwater surface" with "groundwater table" and "below surface" to "groundwater depth below land surface".

*P7: I would mention that groundwater levels were recorded with submersible pressure transducers.*

20   – Yes, technique for groundwater level monitoring added to revised manuscript in the Groundwater section.

*P7 L7: a screen interval to 32 m depth would place it below the 30-m thick Pampeano formation (P4 L10). Can you state that the wells didn't penetrate the Puelche formation, or that the groundwater levels are a composite of the two formations? If the intent is to measure gw levels in the Pampeano, its surprising they would be screened with such a long interval, and so close to the bottom of the formation.*

25   – By continuous inspection of the drill pads, it was carefully surveyed and confirmed during well drilling that the drilling stopped within the clay layer that overlays the Puelche formation. Thus, the monitored groundwater levels represent exclusively the Pampeano aquifer.

*P7 L13: Its unclear what p is here and elsewhere. The p-value from a statistical test?*

– Yes, this is the p-value of statistical testing. Corrected/clarified in the revised manuscript.

30  *P8 L8: These SY values appear to agree very well with the gravity data, based on figure 4. At some point it would be interesting to compare those estimates, not necessarily in this paper. But you could mention the good agreement (some readers may not realize gravity data are useful for estimating SY).*

– This is a very good point. A study that assesses the value of the gravity observations for specific yield estimation is currently in preparation by co-authors of this manuscript. Without going into further details here, we add the following
35    sentence: "As discussed and shown in section 3.3.3 and Figure 4, these estimations of specific yield are in good agreement with gravity residuals, underlining the value of gravity observations for hydrogeological studies."

*P8 L17: delete "of" (1.7% missing data).*

– Corrected.

*P8, L18: I only found data through March 18 in IGETS. I assume it will be updated at some point? Assuming this is a long-term site, can these data sets (from this paper and IGETS) be maintained/updated "automatically"?*

– The AGGO site is indeed a long-term site. The SG time series are processed and uploaded to IGETS by the official provider irregularly after exploratory analysis (hindering automatic upload). For this study, we had a direct access to the SG measurements. The processed series and the corresponding scripts are provided to all users (see Data and Code availability). However, our results are not uploaded to IGETS as these are not the official products provided there.

*P8 L21: Suggest defining "WMO" abbreviation at first use*

– Explained in the revised manuscript (World Meteorological Organization).

*P8 L27: Section 3.3.1 describes gravity residuals, do you mean 3.3.2 and/or 3.3.3?*

– Corrected to 3.3.2.

*P9 L1: trees not tries*

– Corrected.

*P10 L12: I would state explicitly what corrections were applied, e.g. "The data set contains gravity residuals corrected for… as well as…"*

– Revised: "The data set contains gravity residuals corrected for tides, polar motion and length of day effects, local air pressure, and drift. Additional modelled gravity variations that aimed at further correction of the residuals for major environmental effects, such as global atmospheric, oceanic and hydrological mass variations are provided as well".

*P10 L20: Are Level 1, 2, 3 in this paper used the same as at IGETS? That would be worth mentioning in the introduction.*

– Only Level 1 products are identical (input for our processing). The fact that the gravity residuals may differ from official IGETS product is now stated in the revised manuscript: "In this study, only Level 3 hourly gravity residuals are provided. These may differ from IGETS Level 3 products due to different processing strategies."

*P10 L19: Here you discuss gravity time series under the heading "Gravity residuals". Maybe move the mention of Level 1 and level 2 to the general "Gravity" heading?*

– The part discussing the Level 1 and 2 IGETS data is moved to section "Gravity" as suggest by the reviewer. The processing steps necessary for the computation of gravity residuals (including calibration) are described in section "Gravity residuals".

*P10 L 21: "In this study, only Level 3 hourly gravity residuals are provided": unclear. Do you mean, gravity residuals are only provided as a level 3 product? (Do IGETS Level 2 products include residuals?)*

– This part is revised to clarify the topic following the referee's comment: "The IGETS database provides Level 2 products (series corrected for instrumental issues ready for tidal analysis) processed either by the station operator or at the University of French Polynesia."… "In this study, only Level 3 hourly gravity residuals are provided. These may differ from IGETS Level 3 products due to different processing strategies"

*P10 L 22: I would be interested to learn how the calibration factor was determined.*

– The following explanation is added to the revised manuscript: "These parameters were estimated by using co-located absolute gravity measurements carried out with a FG-5 gravimeter (calibration factor) and by evaluating the system response to an injected step function (phase shift)."

*P12 L8: You could mention storm surges here as a major contributor to the non-tidal ocean loading – it took me a while (and the Oreiro paper) before I figure out what this was.*

– Added to the revised manuscript: "As shown in Oreiro et al. (2018), the effect of non-tidal ocean loading by storm surges plays a very important role for gravity recordings at AGGO. In this study, the corresponding gravity effect was computed using four models with global coverage..."

*P12 L10: what exactly is the hydrological effect? (soil moisture + groundwater + precip + ET?) Its surprising daily rainfall would suffice for hourly residuals.*

– In the section the reviewer is referring to, we only describe the large-scale hydrological effect. The following extension is accordingly added to the revised manuscript: "The gravity effects were computed for an integration radius larger than $0.1°$, using all water storage compartments that were given by the individual models, mainly soil moisture up to a model-specific soil depth, and snow storage". Only the state variables of water storage as an expression of hydrological mass changes are taken into account here, no fluxes such as precipitation or ET.

*P12 L 30: Add "m" after 0.1*

– Corrected.

*P13: I would mention specifically that code is provided as Matlab, Julia, and shell scripts (+ others?), and that the relational database is SQL. For what it's worth, I was unable to follow the instructions for using MySQL (I don't have any experience using it). I was able to create a database and run the commands in create_hosgo_db.sql and fill_hosgo_db_metadata.sql from the SQL command prompt, but I got several errors trying to run the commands in fill_hosgo_db_data.sql, all of the form: ERROR 1452 (23000): Cannot add or update a child row: a foreign key constraint fails ('hosgo'.'timeseries', CONSTRAINT 'timeseries_ibfk_1' FOREIGN KEY ('ts_id') REFERENCES 'timeseries_info' ('ts_id'))*

– The suggested specification has been added: "The repository contains a set of example commands in MySQL. The processing scripts are written in Julia and Matlab programming languages."

– Könnte Marvin versuchen die Database zu installieren? Bei mir leuft alles normal.

*Figure 1: Label elevation scale bar in meters. A satellite image in part (c) would be useful.*

– Meter units have been added to the scale bar. As addressed in a previous comment, coordinates of the site that allow the reader to look up satellite images are now included in the figure caption. The main reason for not including a satellite image itself is the often unclear license conditions.

*Figure 2: Be more specific about groundwater units, both in the y-axis label and the caption. If you are reporting negative values, it is probably groundwater elevation relative to land surface. More typical would be "Depth below land surface", with positive values and a reverse y-axis, or elevation relative to mean sea level, also in positive values.*

– Figure now with positive values and reversed y-axis.

– Modified Figure caption: "groundwater depth below land surface in $\mathrm{m}$"

*Figure 4: Perhaps outside the scope of the paper, but I would be interested to see additional time series: the gravity effect of soil moisture, goundwater, air pressure, local loading, global loading, etc., plus the gravity residuals before applying air pressure and hydromet corrections. It appears you've simulated the residuals nearly exactly from the hydromet data and models; what does the residual look like after that correction – it must be nearly flat? What signal(s) might you see in such a time series?*

– The residual signal is subject to a study currently conducted by co-authors of this manuscript. It's presentation and discussion would exceed the scope of this data publication. Nonetheless, please find in the following a figure (Figure 1) showing the residuals as described by the referee. The plot shows the residuals corrected for all available global and local effects and using the large-scale NTOL effect as estimated by OMCT RL6 (in blue) and the regional Estuary model (black). It should be noted that the final NTOL should comprise both, large-scale and the regional effects. The figure is in the same scale as the original used in the manuscript to highlight the significant reduction of the variation after corrections.

[Figure]

**Figure 1.** Gravity residuals corrected for all available global and local effects and using the large-scale NTOL effect as estimated by OMCT RL6 (in blue) and the regional Estuary model (black).

–

*References: There appears to be a formatting error in which the URL is duplicated (with slight changes) for many of the references*

– This issue is resolved in the revised version.

5   Reply to "**RC2 by Anonymous Referee 2**"

We thank the referee for his very positive overall evaluation of the manuscript. Here are our answers to his/her specific comments:

10  *In the following I would like to note a few minor details. (1) First there are slight redundancies in the representation of the 3 data levels (page 2, line 30 and following) and p.5, L 2 and following.*

– We agree that there is some slight repetition in describing the data types, but because on page 2 (at the end of the introduction chapter) we give a summary description of the three data levels as an overview while a more detailed description of the levels including specific technical processing steps is given in the Data section on page 5, we decided
15      to keep this twofold but overall differing description.

*(2) Please do not use cgs units but SI units (p. 7 table 2)*

– Corrected in the revised version.

*(3) The accuracy of the percentages (in the first column in Table 2) allows a number representation up to the second decimal?*

– No, this would exceed the accuracy in this case. Corrected to one decimal.

20  *(4) In general, I find the spatial relationship between illustrations and description in the text to be too large. Both should be presented more in relation to each other. The same applies to the tables (Table 4 and Section 3.3.2).*

– We agree, this will carefully be considered in the layout settings of the final publication.

*Figures Please, show in fig. 1a the position of the cities of La Plata and Bs. Aires.*

– Done in revised version.

*Enlarge fig. 1b and replace the yellow colour with a different one – it is hard to read. Explain "prec." "meteo", SM (??), SLR, GNSS etc. I suggest to include a photo showing some parts of the interior – if possible.*

– Colour replaced, and explanations to the abbreviations added to the figure caption. We decided not to include photos to save space and because of their limited information content given that available photos do not show much more than the instrument such as the superconducting gravimeter itself. This is available from other sources, too.

*All other pictures are too small for my opinion - enlarge, if possible.*

– Done in revised version, extended to full width of page.

Reply to "**RC3 by Jeff Freymueller**"

*My comments are limited to minor corrections, as shown in the annotated manuscript. The data set looks to be complete and useful, and the descriptions are comprehensive.*

– Thanks for the positive evaluation. Minor corrections are considered in the revised manuscript as suggested.